# Thymic Exhaustion and Increased Immune Activation Are the Main Mechanisms Involved in Impaired Immunological Recovery of HIV-Positive Patients under ART

**DOI:** 10.3390/v15020440

**Published:** 2023-02-05

**Authors:** Maria Carolina Dos Santos Guedes, Wlisses Henrique Veloso Carvalho-Silva, José Leandro Andrade-Santos, Maria Carolina Accioly Brelaz-de-Castro, Fabrício Oliveira Souto, Rafael Lima Guimarães

**Affiliations:** 1Keizo Asami Institute (iLIKA), Federal University of Pernambuco—UFPE, Recife 50670-901, PE, Brazil; 2Department of Genetics, Federal University of Pernambuco—UFPE, Recife 50670-901, PE, Brazil; 3Aggeu Magalhães Institute (IAM/FIOCRUZ), Recife 50740-465, PE, Brazil; 4Vitória Academic Center (CAV), Federal University of Pernambuco—UFPE, Recife 55608-680, PE, Brazil; 5Agreste Academic Center (CAA), Federal University of Pernambuco—UFPE, Recife 55014-900, PE, Brazil

**Keywords:** AIDS, immunological non-responders, antiretroviral therapy, CD4+ T cell reconstitution, immune activation

## Abstract

Decades of studies in antiretroviral therapy (ART) have passed, and the mechanisms that determine impaired immunological recovery in HIV-positive patients receiving ART have not been completely elucidated yet. Thus, T-lymphocytes immunophenotyping and cytokines levels were analyzed in 44 ART-treated HIV-positive patients who had a prolonged undetectable plasma viral load. The patients were classified as immunological non-responders (INR = 13) and immunological responders (IR = 31), according to their CD4+ T cell levels. Evaluating pre-CD4+ levels, we observed a statistically significant trend between lower CD4+ T cell levels and INR status (Z = 3.486, *p* < 0.001), and during 18 months of ART, the CD4+ T cell levels maintained statistical differences between the INR and IR groups (WTS = 37.252, *p* < 0.001). Furthermore, the INRs were associated with an elevated age at ART start; a lower pre-treatment CD4+ T cell count and a percentage that remained low even after 18 months of ART; lower levels of recent thymic emigrant (RTE) CD4+ T cell (CD45RA + CD31+) and a naïve CD4+ T cell (CD45RA + CD62L+); higher levels of central memory CD4+ T cells (CD45RA-CD62L+); and higher immune activation by CD4+ expressing HLA-DR+ or both (HLA-DR+ and CD38+) when compared with IRs. Our study demonstrates that thymic exhaustion and increased immune activation are two mechanisms substantially implicated in the impaired immune recovery of ART-treated HIV patients.

## 1. Introduction

Currently, more than 38 million people are living with HIV-1 (human immunodeficiency type 1) around the world [1]. Over the past four decades, since the beginning of the pandemic, many advances have been made, especially regarding antiretroviral therapy (ART), whose regimens are based on many different available antiretrovirals [2,3]. ART aims to reduce plasma viral load to undetectable levels (<40 RNA copies/mL), making HIV-positive individuals less susceptible to developing opportunistic infections and neoplasms, and decreasing the number of AIDS-related deaths. Presently, more than 28 million people with HIV use ART [4,5].

It is expected that, with the course of treatment, the loss of CD4+ T cell levels during HIV infection may be gradually recovered. However, 15–30% of ART-treated patients, despite complete viral suppression, fail to recover immunologically [6]. These patients are defined as virological responders but immunological non-responders (INR), and they are more susceptible to HIV-related complications and an increased risk of death [7,8,9].

Although there are no clear and well-established mechanisms, impaired immunological recovery has been described as a multifactorial condition, and factors such as advanced age, male sex, low pre-treatment CD4+ T cell count, residual viral replication, genetic factors, and coinfections during ART may influence CD4+ T cell restoration [10,11,12,13]. Moreover, some studies have demonstrated that dysregulation of T lymphocyte homeostasis is a determining aspect of immune reconstitution in ART-treated patients. It is regulated by a dynamic balance between CD4+ T cell production, destruction, and trafficking to and from lymphoid organs and peripheral circulation [6,7].

Two main processes have been associated with the imbalance in T cell homeostasis and, consequently, with the impaired immunological recovery: the decrease in thymic production of T lymphocytes and the excessive destruction of these cells [8,14]. Thymic function can be evaluated, among others, by the output of RTE (recent thymic emigrants) and naïve CD4+ T cells [15,16,17]. The reduction of these cells has been reported as a contributing factor to ineffective immune recovery [18,19,20]. In addition, cytokine production, which normally results from cell death, has been investigated to increase immune activation, another important mechanism involved in poor immune reconstitution. This process can generate the trigger of a vicious cycle of cell death and reduce CD4+ T cell count [21,22,23].

Studies that aim to provide more data regarding immunological recovery deficiency are essential to better understand this condition due to its complexity in an HIV infection context. Thus, the present study used immunophenotypic tools to investigate different aspects of thymic function, immune activation, T lymphocytes, and cytokine profiles, seeking to clarify the impaired immunological recovery in ART-treated HIV-positive patients.

## 2. Materials and Methods

### 2.1. Study Population

This study consisted of 44 HIV-positive patients (33 males and 11 females) under ART recruited from the Instituto de Medicina Integral Professor Fernando Figueira (IMIP), Pernambuco state (Northeast Brazil), between 2018 and 2020. The general inclusion criteria were age over 18 years old, undergoing ART for at least 18 months with a prolonged undetectable viral load (<40 copies/mL), and good adherence to treatment. The exclusion criteria were pregnancy, autoimmune diseases, cancer, and a history of injecting drug use. Sociodemographic and clinical data were collected from medical records: age at the ART start date, time until ART start after HIV diagnosis, ART regimens, pre- and during-treatment viral loads as well as CD4+ and CD8+ T cell counts and percentages, and serological data regarding co-infections (hepatitis B virus—HBV, hepatitis C virus—HCV, syphilis, cytomegalovirus—CMV, toxoplasmosis, herpes simplex virus types 1 and 2—HSV-1/2, and human T cell lymphotropic virus types I and II—HTLV-I/II). All patients answered standard questionnaires and signed written informed consent forms, providing blood samples for immunological analyses.

### 2.2. Immunological Classification

These 44 HIV-positive ART-treated patients, who had undetectable plasma viral loads (<40 copies/mL) during the first 18 months of ART and maintained them at all clinical appointments, were classified according to CD4+ T cell count recovery. The classification is based on a late or early ART start after HIV diagnosis, defined by the 500 CD4+ T lymphocytes (cells/μL) threshold. Therefore, patients who start ART with a CD4+ T cell count ≥500 cells/μL or achieve a CD4+ T cell count above that threshold after 18 months of ART are classified as immunological responders (IR). The other patients, who started ART with a CD4+ T cell count <500 cells/μL, and even after 18 months of ART maintained the CD4+ T cell levels below that threshold, were classified as INR [6]. Thus, 13 ART-treated patients (11 males and 2 females) were classified as INR, and the remaining 31 patients (22 males and 9 females) were defined as IR.

### 2.3. Plasma and PBMC Isolation

A total of 4 mL of blood was collected from all ART-treated patients in EDTA tubes. Plasma was separated in cryovials with a protease inhibitor and stored at −80 °C until the cytokine measurement. Peripheral blood mononuclear cells (PBMC) were separated by Ficoll-Paque Plus density gradient centrifugation and washed twice with PBS 1x followed by resuspension in FACS buffer (3% BSA, 0.01% NaN3—sodium azide, PBS 1×). Cell viability (>95% on average) was determined by the Trypan blue (0.4%) exclusion test.

### 2.4. Immunophenotyping

Combinations of different fluorescent monoclonal antibodies were used to stain PBMCs for 20 min at room temperature, protected from light. Then, PBMCs were washed with FACS buffer, fixed with PBS-formaldehyde 1%, and analyzed by flow cytometry using a FACSAria III cytometer (BD Biosciences). For each sample, 100,000 events were acquired and gated (Appendix A) to detect the following T lymphocyte subsets as follows: activated CD4+ T cells (CD4+ CD38+ HLA-DR+); activated CD8+ T cells (CD8+ CD38+ HLA-DR+); RTE CD4+ T cells (CD4+ CD45RA+ CD31+); naïve CD4+ T cells (CD4+ CD45RA+ CD62L+); central memory CD4+ T cells (CD4+ CD45RA- CD62L+); effector memory CD4+ T cells (CD4+ CD45RA- CD62L-); and effector CD4+ T cells (CD4+ CD45RA+ CD62L-). Immunofluorescent monoclonal antibodies APC-CD4 (EDU-2), BB515-CD4 (RPA-T4), PE-CD31 (WM59), PercpCy5.5-CD45RA (HI100), APC-CD62L (DREG-56), PECy7-CD8 (RPA-T8), PE-CD38 (HIT2), and FITC-HLA-DR (G46-6) were obtained from BD Biosciences and BioAlbra. The acquired data were analyzed using the FlowJo software, version 10.

### 2.5. Cytokines Measurement

Plasma IL-2, IL-6, IL-10, and TNF-α protein levels were measured by the Cytometric Bead Array (CBA) Human Th1/Th2 Cytokine Kit II and the manufacturer’s instructions (BD Biosciences) were followed by using an Accuri C6 cytometer (BD Biosciences). CBA data were analyzed using the FCAP Array software.

### 2.6. Statistical Analyses

Categorical (sociodemographic and clinical) variables were evaluated by Fisher’s exact test for association with INR status. The Shapiro–Wilk test was used to check the normal distribution. Therefore, variables that followed a normal distribution were compared among groups by Student’s *t*-test for independence and displayed as mean ± SD. The Mann–Whitney test was used to compare the groups for variables that did not follow a normal distribution, which displayed values as the median and interquartile range (IQR). The Cochran-Armitage test was performed to assess the trend in HIV-infected patients’ distribution between the INR and IR groups according to their pre-treatment CD4+ T cell count. To analyze CD4+ T cell count over ART time between the groups, the Wald-type test was used, and then a Wilcoxon signed-rank test adjusted by Bonferroni was performed for post hoc pairwise comparisons. The statistical significance level (α) was set at 0.05 for all tests. All statistical analyses were carried out using the R program (version 4.2.0) and GraphPad Prism (version 8.0.1).

## 3. Results

### 3.1. Clinical Data Analyses

We observed that age at ART start showed a significant difference between the two groups (*p* = 0.041), with means of 41.8 ± 3.0 years old for the INR group and 33.7 ± 2.1 for the IR group. We also analyzed sex, body mass, pre-treatment PVL (plasma viral load), time to start ART post-diagnosis, ART regimens, and their changes. However, there were no statistical differences among the groups for all these variables (Table 1).

Regarding pre-treatment CD4+ T cell count, there was a statistically significant difference between the two groups. INR group showed a pre-treatment CD4+ T cell count average of 153.9 cells/µL, while the IR group showed 554.5 cells/µL (*p* < 0.001), demonstrating that IRs started ART with higher CD4+ T cell levels. The same could be observed concerning the pre-treatment CD4/CD8 ratio (*p* = 0.010) and pre-treatment CD4% (*p* < 0.001), which were significantly higher in the IR group. Additionally, we also demonstrated that there was a statistically significant difference in pre-treatment CD8 percentages among the groups (INR: 65.4 ± 8.4 and IR: 47.1 ± 4.8; *p* = 0.008). This statistical pattern was maintained after 18 months of ART, where the CD4+ T cell count (*p* < 0.001), the CD4/CD8 ratio (*p* < 0.001), and CD4% (*p* < 0.001) continued significantly higher in the IR group, while CD8% (*p =* 0.001) remained higher in the INR group, having a significant influence on immunological recovery during ART (see Table 1 for more details).

Corroborating with the data found about the CD4+ T lymphocyte count, we observed that the lower the pre-treatment in CD4+ T cell levels, the greater the trend of ART-treated patients being INRs (Z = 3.486; *p* < 0.001) (Figure 1). Most INRs in our study (>60%) showed a CD4+ T lymphocyte count ≤200 cells/µL in the pre-ART period, while there were few patients (<10%) in the IR group with this pre-ART CD4+ T cell count. In addition, the pairwise comparison analyses with CD4+ T cell count along therapy time (pre-ART, 6 m of ART, 12 m of ART, and 18 m of ART) showed a significant difference in CD4+ T cell gain between the IR and INR groups (WTS = 37.252; *p* < 0.001) (Figure 2). CD4+ T lymphocyte count recovery was higher in the IR group compared to the INR group, especially in the last 6 months of ART, where the difference among the groups was greater. It is possible to observe that after 12 months of therapy, the IR group continued increasing CD4+ T lymphocyte levels, while in the INR group there was a decline in this process, suggesting an exhaustion in CD4+ T cell gain (Figure 2).

In this study, the coinfections did not influence immunological recovery in our population (see Appendix A for more details).

### 3.2. Thymic Output and T Cell Subsets

The thymic function analysis was performed by quantifying RTE CD4+ T cells (CD4+ CD45RA+ CD31+, Figure 3A) and naïve CD4+ T cells (CD4+/CD45RA+ CD62L+, Figure 4C). RTE percentage was lower in the INR group (19.5 ± 6.3) compared to the IR group (29.9 ± 11.5) with a statistically significant difference (*p* = 0.038, Figure 3B), demonstrating the influence of these cells on immune reconstitution.

The ART-treated groups were also analyzed regarding CD4+ T cell subsets through the CD45RA and CD62L markers (Figure 4A), consisting of central memory CD4+ T cells (T_CM_ = 40.1%), naïve CD4+ T cells (T_N_ = 35.2%), effector memory CD4+ T cells (T_EM_ = 20.3%), and effector CD4+ T cells (T_EFF_ = 3.2%). A statistically significant difference was found between the two groups regarding T_CM_ cells. For this analysis, the INR group showed a higher T_CM_ percentage than the IR group (IR = 34.4 ± 8.9% vs. INR = 45.9 ± 14.5%; *p* = 0.028, Figure 4B). Furthermore, it was also possible to observe that the T_N_ fraction percentage was significantly lower in the INR group (IR = 44.3 ± 15.0% vs. INR = 26.1 ± 7.7%; *p* = 0.007; Figure 4C). This result corroborates with the RTE CD4+ T cell %, which was also lower in the same group, suggesting a reduced thymic function in INR patients. No significant differences were found between the groups for the other CD4+ T cell subsets (Figure 4D,E).

### 3.3. Immune Activation

The analysis of the immune activation profile (HLA-DR+ CD38+, Figure 5A) showed an increased activation profile in the INR group compared to the IR group, mainly through the expression of HLA-DR (IR = 4.9 ± 2.7 vs. INR = 12.9 ± 6.2; *p* = 0.007) and both markers (IR = 1.1 ± 0.5 vs. INR = 2.6 ± 1.1; *p* = 0.004) on the surface of the CD4+ T cells (Figure 5B). Although the CD8+ T cells % was statistically different among the ART-treated groups (IR = 34.5 vs. INR = 46.4, *p* = 0.001, Table 1), a similar percentage of activated CD8+ T cells was found between the INR and IR groups with reduced activation levels of CD8+ T cells (Figure 5C).

### 3.4. Plasma Cytokines Levels

We also performed the quantification of the plasma levels of pro-inflammatory and anti-inflammatory cytokines (TNF-α, IL-2, IL-6, IL-10) to evaluate the Th1 and Th2 profiles, respectively. Regarding that, no statistical differences were found between the INR and IR groups for these proteins (Figure 6A–D), showing similar cytokine production during ART.

## 4. Discussion

The reconstitution of CD4+ T lymphocytes in patients undergoing ART occurs gradually and persistently over the years, providing a better quality of life to these HIV-infected individuals [7]. However, despite the efficacy of ART, some individuals, even after achieving suppressed viral replication for a long time, maintain a low concentration of CD4+ T cells. Consequently, these individuals remain susceptible to HIV-related complications and death [6,24]. Therefore, the present study evaluated clinical factors involved in immune recovery, as well as thymic function, immune activation, T lymphocytes, and cytokine profiles, to analyze their influence on CD4+ T cell reconstitution in ART-treated patients.

Some factors have been reported to influence immunological recovery during ART, mainly the pre-treatment CD4+ T cell count [25,26]. We observed that low CD4+ T lymphocyte levels at the beginning of ART were associated with poor immune reconstitution, corroborating with other studies [27,28,29]. In our analysis, most individuals in the INR group had a pre-treatment CD4+ T lymphocyte count ≤200 cells/µL, demonstrating that they gained fewer CD4+ T cells, which persisted lower than IRs over ART time, even after complete and prolonged PVL suppression, suggesting some exhaustion in the immune system. These results are supported by the fact that individuals who start ART with CD4+ T cell baseline ≥500 cells/µL have a decreased risk of AIDS progression and a higher probability of maintaining viral suppression for a long time and achieving adequate immunological recovery when compared to those with pre-CD4+ T cell counts ≤200 cells/µL [30,31,32,33]. Furthermore, immune reconstitution is less evident in HIV-positive patients who started ART with reduced CD4+ T cell levels [20,34].

The lower output of RTE cells (CD45 + CD31+%) and naïve CD4+ T cells (CD45RA+CD62L+%) found in the INR group can be explained by the thymic exhaustion presented in these patients. During ART, the greatest increase in CD4+ T cell count occurs in the first six months of therapy, as demonstrated in our analysis and corroborated by other studies [35,36,37]. The following phases consist of slightly slower gains and are mainly the result of thymopoiesis [7]. It is known that the thymus presents maximum activity during childhood, generating a wide repertoire of T lymphocytes that would be sufficient throughout life. With aging, the functional thymic tissue is replaced by adipose tissue, and the organ undergoes an involution process that results in decreased T cell production and maturation [38,39]. However, under massive T cell depletion, the thymic function is again required in adulthood, such as in HIV infection [16,40]. HIV can infect, among other cells, thymocytes and, consequently, affect the process of thymopoiesis [41]. Thus, long periods of HIV infection without ART can lead to thymic exhaustion and inefficient immune recovery. This fact demonstrates the importance of early treatment initiation, as recommended by the WHO [42]. Furthermore, thymic exhaustion is also related to exacerbated immune activation since the HIV-induced proliferation of activated thymocytes is capable of promoting inflammatory activity that can cause increased immune activation and damage to functional tissue [43,44].

In this context, we also observed an association of age at the ART beginning with immunological recovery failure, corroborating with other authors [45,46]. In our study, the INR group consisted of older patients than those in the IR group. They showed lower thymic output, which may be a result of late diagnosis and initiation of treatment and immunosenescence, a physiological process that occurs naturally with aging [47,48]. Moreover, aging is also linked to increased immune activation, another important mechanism involved in impaired immune reconstitution [7,49,50].

Exacerbated immune activation is a predictor of disease progression as effectively as the plasma viral load, and it has been associated with persistently reduced CD4+ T cell levels during ART [34,51]. Our study also demonstrated that there were higher levels of immune activation markers, mainly HLA-DR, on CD4+ T lymphocytes’ surfaces in the INR group compared to the IR group. Naturally, HIV-1-infected individuals have persistent immune activation and inflammation, especially when they are not under treatment [52,53]. In the presence of ART and its consequent reduction in viral load, the activation of CD8+ T cells decreases [54,55], and it is possible to observe the same pattern in both groups. However, the increased immune activation found in INRs may be a result of high cell death levels. As previously demonstrated by our group in a similar population, the INR group showed elevated peripheral CD4+ T cell death by pyroptosis compared to the IR group [18]. Pyroptosis is a highly inflammatory cell death via caspase-1, which promotes the release of pro-inflammatory cytokines such as IL-1β and IL-18 [56]. Together, these molecules are responsible for attracting more cells to death, resulting in a vicious cycle of cell depletion [57,58]. IL-1β also plays an important role by activating memory T cells and stimulating their immune functions [59,60]. This may explain the percentage difference found between the INR and IR groups regarding central memory CD4+ T cells in our study.

Other cytokines play an important role in HIV-1 infection. The balance of pro- and anti-inflammatory cytokine production is responsible for influencing levels of viral replication and the homeostatic balance of T cells [61]. In the context of immune recovery, some studies have reported that patients with lower T cell counts could benefit from increased IL-2, promoting cell proliferation events [62]. In addition, high levels of IL-6, TNF-α, and IL-10 found in these patients can affect T cell exhaustion, proliferation, and immune activation [63,64,65]. However, in our study, it was not possible to observe statistically significant differences in the cytokine levels evaluated between the groups.

In addition to the factors already mentioned, different therapeutic regimens can also influence immune recovery. Some studies have reported faster viral suppression and better CD4+ T cell gain in patients whose ART regimens contain dolutegravir (DTG, an integrase inhibitor) instead of efavirenz (EFZ, a non-nucleoside reverse transcriptase inhibitor) [4,66]. In our study, most of the ART-treated patients (79.5%) used the first-line regimen recommended by the World Health Organization (WHO) [42,67] (TDF + 3TC + EFZ or DTG), and there was no statistical difference regarding immunological recovery.

In conclusion, our data propose that reduced thymic function and increased immune activation play a key role in poor CD4+ T lymphocyte reconstitution. These results help to better understand the impaired immunological recovery in HIV-positive patients under ART. Moreover, it could provide data that may contribute to the development of new therapeutic strategies, focusing on decreasing immune activation and increasing thymic function, providing a better outcome for these patients, known as INRs.

## Figures and Tables

**Figure 1 viruses-15-00440-f001:**
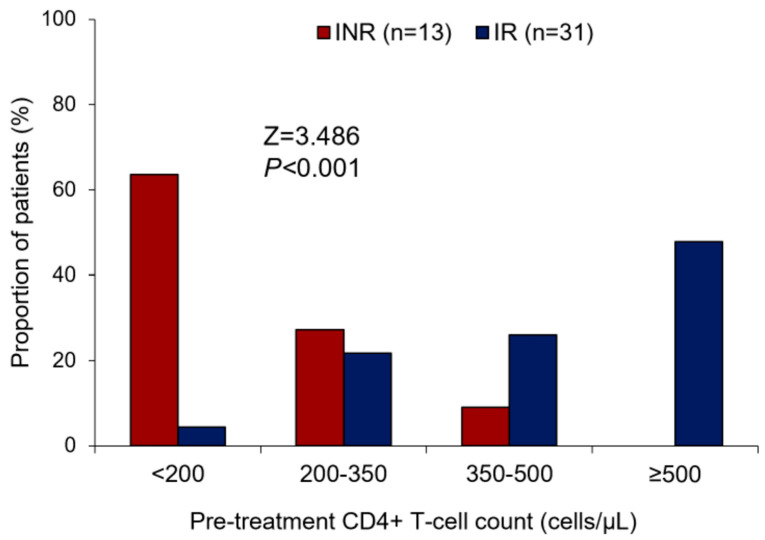
Bar plot showing the comparison of the INR and IR groups about the pre-treatment of CD4+ T cell count stratification. Cochran–Armitage test for trend: Z = 3.486; *p* < 0.001. INR—immunological non-responders. IR—immunological responders.

**Figure 2 viruses-15-00440-f002:**
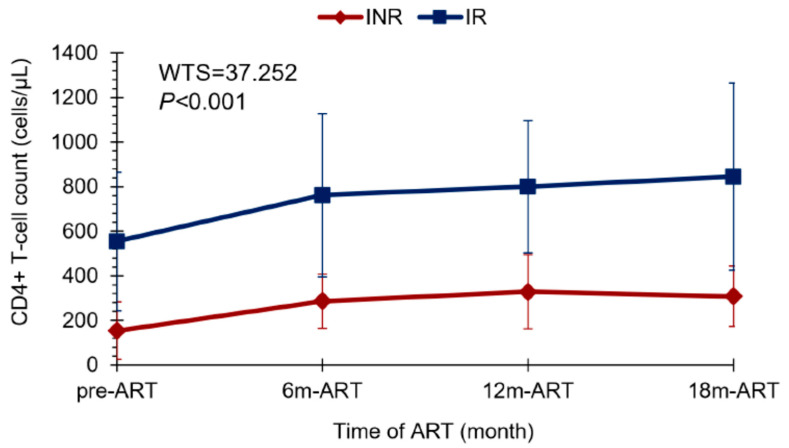
Line plot showing the CD4+ T cell gain profile between the INR and IR groups throughout ART time: pre-treatment, at 6 months, 12 months, and 18 months of ART. Wald-type test: WTS = 37.252; *p* < 0.001. ART—antiretroviral therapy. INR—immunological non-responders. IR—immunological responders.

**Figure 3 viruses-15-00440-f003:**
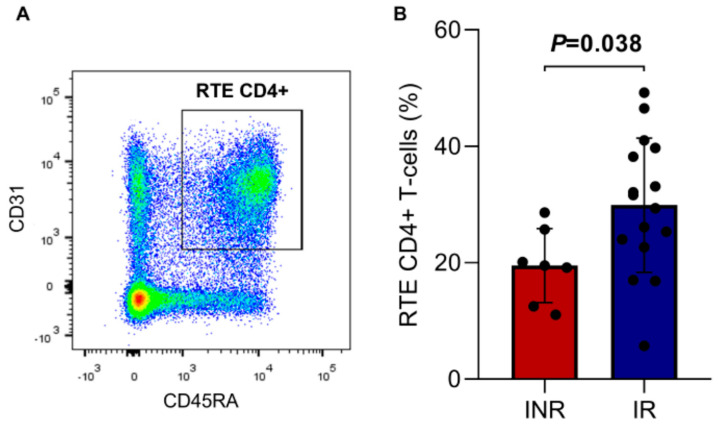
(**A**) Representative dot plot showing the gating strategy for RTE CD4+ T cells (CD4+/CD45RA+CD31+). (**B**) Percentage of RTE CD4+ T cells in the INR and IR groups. Means, standard deviation, and *p*-value (according to the *t*-test) are shown. INR—immunological non-responders. IR—immunological responders. RTE—recent thymic emigrants.

**Figure 4 viruses-15-00440-f004:**
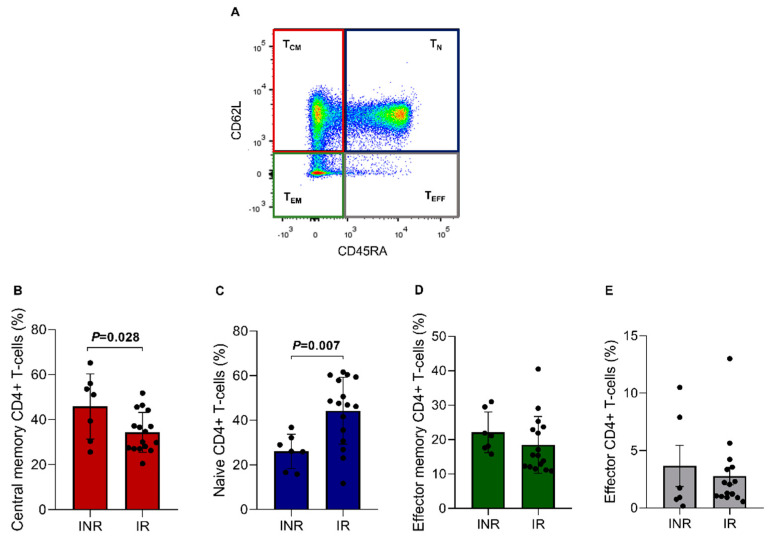
CD4+ T lymphocytes subsets in the INR and IR groups. (**A**) Lymphocytes were gated on CD4+ T cells and then on CD45RA and CD62. (**B**) *T*_CM_ = central memory CD4+ T cells (CD45RA- CD62L+); (**C**) *T*_N_ = naïve CD4+ T cells (CD45RA+ CD62L+); (**D**) *T*_EM_ = effector memory CD4+ T cells (CD45RA- CD62L-); and (**E**) T_EFF_ = effector CD4+ T cells (CD45RA+ CD62L-). Means, standard deviation, and *p*-values (according to the *t*-test) are shown. INR—immunological non-responders. IR—immunological responders.

**Figure 5 viruses-15-00440-f005:**
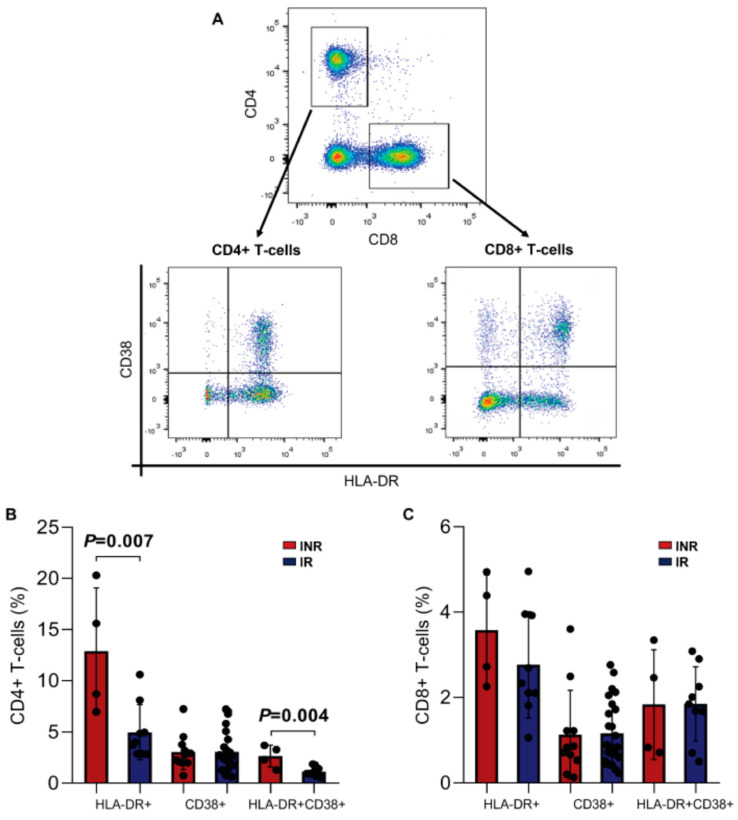
Immune activation of CD4+ and CD8+ T cells in the INR and IR groups. (**A**) Representative dot plots showing the gating strategy for activated CD4+ (CD4+/CD38+ and/or HLA-DR+) and CD8+ (CD8+/CD38+ and/or HLA-DR+) T-lymphocytes. (**B**) Percentage of HLA-DR+, CD38+, and both in CD4+ T cells. (**C**) Percentage of HLA-DR+, CD38+, and both in CD8+ T cells. Means, standard deviation, and *p*-values (according to the *t*-test) are shown. INR—immunological non-responders. IR—immunological responders.

**Figure 6 viruses-15-00440-f006:**
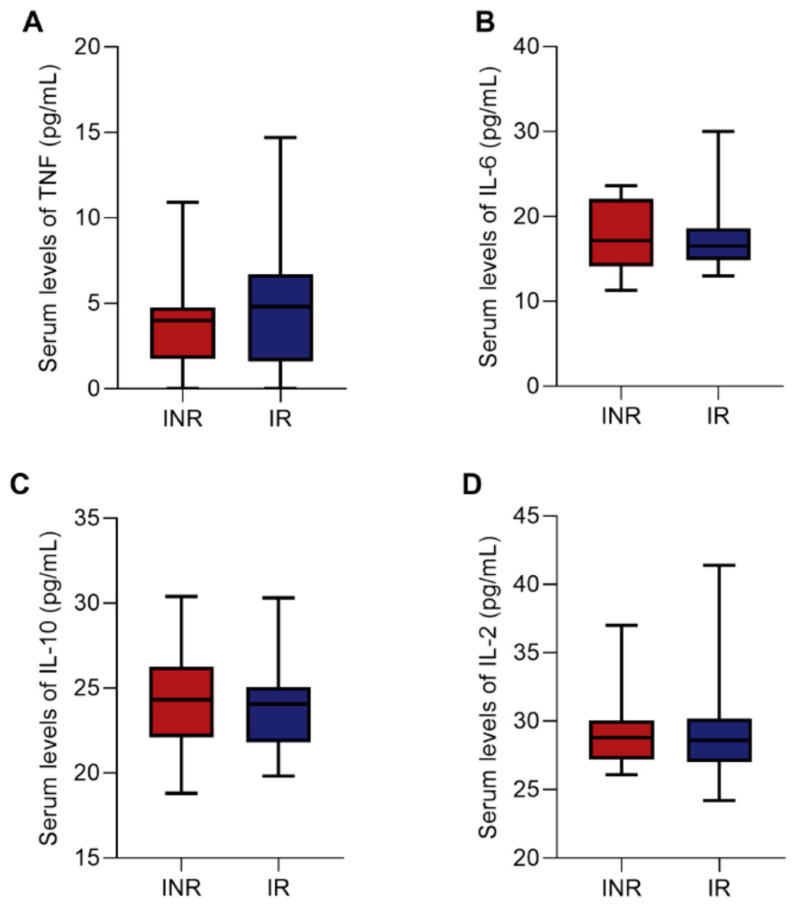
Box plots showing the plasma levels of TNF (**A**), IL-6 (**B**), IL-10 (**C**), and IL-2 (**D**) cytokines between the INR and IR groups. Medians and interquartile range are shown. Mann–Whitney test was used to compare INR and IR groups. INR—immunological non-responders. IR—immunological responders.

**Table 1 viruses-15-00440-t001:** Clinical characteristics of HIV-positive patients according to their immunological response during ART.

Variables		INR *n* = 13 (%)	IR *n* = 31 (%)	*p*
Sex	Male	11 (84.6)	22 (71.0)	0.461 ^a^
	Female	2 (15.4)	9 (29.0)	
Age (years old) at ART start date *		41.8 ± 3.0	33.7 ± 2.1	**0.041**
Body mass (kg) *		74.6 ± 4.4	75.3 ± 2.5	0.892
Time (weeks) to start ART post-diagnosis **		2.0 (1.0–9.5)	8.0 (3.0–9.0)	0.176
Detailed ART regimens, NRTI + 3TC + (third option)	ABC + 3TC + DTG	1 (7.7)	0 (0.0)	0.571 ^b^
	ABC + 3TC + NVP	0 (0.0)	1 (3.2)	
	TDF + 3TC + ATV/r	2 (15.4))	4 (12.9)	
	TDF + 3TC + LPV/r	0 (0.0)	1 (3.2)	
	TDF + 3TC + DTG	7 (53.8)	14 (45.2)	
	TDF + 3TC + EFZ	3 (23.1)	11 (35.5)	
ART regimens, stratified by classes	2 NRTI + INI	8 (61.5)	14 (45.2)	0.564 ^b^
	2 NRTI + IP/r	2 (15.4)	5 (16.1)	
	2 NRTI + NNRTI	3 (23.1)	12 (38.7)	
ART regimen change ^c^		3 (23.1)	7 (22.6)	1.000 ^a^
Pre-treatment CD4+ T cell count (cells/µL) *		153.9 ± 38.8	554.5 ± 64.9	**<0.001**
Pre-treatment CD8+ T cell count (cells/µL) *		718.5 ± 251.0	842.5 ± 201.4	0.734
CD4+ T cell count after 18 months of ART (cells/µL) *		307.9 ± 37.6	816.6 ± 81.8	**<0.001**
Pre-treatment CD4/CD8 ratio*		0.12 ± 0.05	0.60 ± 0.10	**0.010**
CD4/CD8 ratio after 18 months of ART *		0.48 ± 0.08	1.34 ± 0.13	**<0.001**
Pre-CD4+ (%) *		9.9 ± 2.2	25.0 ± 1.7	**<0.001**
Pre-CD8+ (%) *		65.4 ± 8.4	47.1 ± 4.8	**0.008**
CD4+ (%) after 18 months of ART *		19.5 ± 2.1	38.2 ± 1.6	**<0.001**
CD8+ (%) after 18 months of ART *		46.4 ± 3.4	34.4 ± 1.8	**0.001**
Pre-treatment PVL (log_10_ RNA copies/mL) *		4.7 ± 0.4	4.1 ± 0.2	0.129

* *t*-test (Shapiro–Wilk test: >0.05), values displayed as mean ± SE. ** Wilcoxon–Mann–Whitney test (Shapiro–Wilk: <0.05), values displayed as median (IQR). ^a^ Fisher’s exact test. ^b^ Chi-squared test. ^c^ Change from a NNTRI-containing cART (EFZ) to a PI/r (ATV/r), INI (DTG), or another NNTRI (NVP)-containing ART regimen. 3TC—lamivudine; ABC—abacavir; ART—antiretroviral therapy; ATV/r—ritonavir-boosted atazanavir; AZT—zidovudine; DTG—dolutegravir; EFZ—efavirenz; INI—integrase inhibitor; INR—immunological non-responders; IR—immunological responders; IQR—interquartile range; LPV/r—ritonavir-boosted lopinavir; NNRTI—non–nucleoside reverse transcriptase inhibitor; NRTI—nucleoside reverse transcriptase inhibitor; NVP—nevirapine; PI/r—ritonavir-boosted protease inhibitor; PVL—plasma viral load; SE: standard error; and TDF—tenofovir.

## Data Availability

The data that support the findings of this study are available from the corresponding author upon reasonable request.

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
