# Peer review of "Thymic Exhaustion and Increased Immune Activation Are the Main Mechanisms Involved in Impaired Immunological Recovery of HIV-Positive Patients under ART"

_viruses, 2023, doi:10.3390/v15020440_

Round 1

Reviewer 1 Report

General Comments

The English language must be thoroughly reviewed.

Authors must use a single definition for the same concept in all the main text, figures and tables. Authors should not use different terms to refer to the same concept. For example: CD4 cell, CD4+ cell, CD4 T cell, CD4+ T cell. Please select the most appropriate and use it throughout the text.

Authors

Lines 5 and 6. The letters d and e to denote the affiliation of the authors should appear in alphabetical order. That is to say, after Castro there must appear c,d and after Souto a,e. Change the name of the Institutions in correspondence to the previous change (lines 11 and 12).

Abstract

The wording of the Abstract must be reviewed since it is difficult to understand. It has sentences that are too long. Please review thoroughly.

Keywords

The keywords should be improved. It is suggested to replace ¨poor T CD4+ cells reconstitution¨ with ¨CD4+ T cell reconstitution¨. The meaning of INR must be written.

Introduction

Please check the wording.

Line 56. The authors must write naїve or naive as they understand, but always in the same way throughout the entire text.

Line 56. Please, delete white space after the CD4+

Materials and Methods

Lines 78, 87, 89, 92, 110-113. CD4+ T cells and CD8+ T cells must appear without the hyphen.

Lines 92-93. Replace CD4-count with CD4+ T cell count

Line 93. Replace immunological non-responders (INR) with INR. That abbreviation is already described in line 47.

Results

Line 141. The authors should define PVL as it is the first time it appears in the text.

Line 159. Please, delete repeated period.  

Line 171. Please, delete white space before de word Most

Line 175. Please, always use the same denomination for the same concept. Substitute CD4 cell for CD4+ T lymphocyte

Line 180. Please, replace CD4+ cell with CD4+ T cell

Line 204. Please, replace RTE CD4% with RTE CD4+ T cells %

Line 225. Please, replace CD8% with CD8+ T cells %

Line 228. Please, replace CD8+ cells with CD8+ T cells

Line 241. Please, delete white space after de word of

Line 249. Authors must specify in the figure caption the statistical test used.

Discussion

Line 257. It is suggested to the authors not to use the term mechanisms since the authors do not evaluate mechanisms but study cell populations.

Line 260. Please, delete white space before de word We

Line 267. Please, delete white space after de word have

Line 271. Please, remove the word cells after the parentheses. It is suggested to write the word cells after RTEs in line 270.

Line 292. It is suggested to replace physiologically with physiological

Line 294. Please, add the period at the end of the sentence.

Line 313. Please, delete white space before de word In

Institutional Review Board Statement

The study was approved by the Ethics Committee of Instituto de Medicina Integral Professor Fernando Figueira in 2013. However, patients were recruited from 2018. Can the authors explain why the study started 5 years after approval? Did this fact influence the quality of the study?

References

1. Line 340. Please review this reference. The title is repeated. The information of the publisher or the web page where it was published must be included.

4. Line 345. Please, delete white space before de word Therapy

6. Line 349. Please, replace CART with cART

7. Line 352. Please, delete white space before de word Therapy

9. Line 357. Please, replace CART with cART

15. Please, review and rectified as correspond the whole reference.

16. Line 377. Please, delete white space after de word Disease

18. Line 383. Please, replace CART with cART

21. Line 391. Please, replace CART with cART

24. Line 400. Please, delete unnecessary white space before doi

37. Line 442. Please, delete unnecessary white space before doi

42. Line 458. Add space after de word HIV. Delete white space

Supplementary Material

Tables, figures, or other supplementary material should appear in the order in which they are cited in the text of the manuscript.

Supplementary Table S1. Please, replace letter b with a

Author Response

Reviewer #1

General Comments

The English language must be thoroughly reviewed.

Response: We appreciate your recommendation, and the manuscript was carefully reviewed by all authors to improve language and grammar.

Authors must use a single definition for the same concept in all the main text, figures and tables. Authors should not use different terms to refer to the same concept. For example: CD4 cell, CD4+ cell, CD4 T cell, CD4+ T cell. Please select the most appropriate and use it throughout the text.

Response: We appreciate your suggestion. “CD4+ T cell” term was used as standard concept throughout the manuscript.

Authors

Lines 5 and 6. The letters d and e to denote the affiliation of the authors should appear in alphabetical order. That is to say, after Castro there must appear c,d and after Souto a,e. Change the name of the Institutions in correspondence to the previous change (lines 11 and 12).

Response: The change has been made.

Abstract

The wording of the Abstract must be reviewed since it is difficult to understand. It has sentences that are too long. Please review thoroughly.

Response: The abstract has been rewritten with shorter and more coherent sentences to make it more understandable. We appreciate your comments.

Keywords

The keywords should be improved. It is suggested to replace ¨poor T CD4+ cells reconstitution¨ with ¨CD4+ T cell reconstitution¨. The meaning of INR must be written.

Response: We appreciate the suggestion. The changes have been made.

Introduction

Please check the wording.

Line 56. The authors must write naїve or naive as they understand, but always in the same way throughout the entire text.

Response: We appreciate your suggestion. The term “naïve” was used as standard.

Line 56. Please, delete white space after the CD4+

Materials and Methods

Lines 78, 87, 89, 92, 110-113. CD4+ T cells and CD8+ T cells must appear without the hyphen.

Lines 92-93. Replace CD4-count with CD4+ T cell count

Line 93. Replace immunological non-responders (INR) with INR. That abbreviation is already described in line 47.

Response: We appreciate the considerations. The replacements were done.

Results

Line 141. The authors should define PVL as it is the first time it appears in the text.

Response: The acronym was defined in the text (line 145). We appreciate the recommendation.

Line 159. Please, delete repeated period.  

Line 171. Please, delete white space before de word Most

Line 175. Please, always use the same denomination for the same concept. Substitute CD4 cell for CD4+ T lymphocyte

Line 180. Please, replace CD4+ cell with CD4+ T cell

Line 204. Please, replace RTE CD4% with RTE CD4+ T cells %

Line 225. Please, replace CD8% with CD8+ T cells %

Line 228. Please, replace CD8+ cells with CD8+ T cells

Line 241. Please, delete white space after de word of

Response: We appreciate your considerations. The replacements were performed, as well as deletion of unnecessary spaces.

Line 249. Authors must specify in the figure caption the statistical test used.

Response: The statistical test used was added in the Figure legend (line 255).

Discussion

Line 257. It is suggested to the authors not to use the term mechanisms since the authors do not evaluate mechanisms but study cell populations.

Response: We appreciate your observation. We changed the sentences to avoid misinterpretation.

Line 260. Please, delete white space before de word We

Line 267. Please, delete white space after de word have

Line 271. Please, remove the word cells after the parentheses. It is suggested to write the word cells after RTEs in line 270.

Line 292. It is suggested to replace physiologically with physiological

Line 294. Please, add the period at the end of the sentence.

Line 313. Please, delete white space before de word In

Response: We appreciate the suggestions. All recommendations were followed.

Institutional Review Board Statement

The study was approved by the Ethics Committee of Instituto de Medicina Integral Professor Fernando Figueira in 2013. However, patients were recruited from 2018. Can the authors explain why the study started 5 years after approval? Did this fact influence the quality of the study?

Response: This study is part of a bigger and ongoing one, which began in 2013 after the approval by IMIP Ethics Committee, and it remains in progress. However, for the analyses presented in this work, biological samples from patients were collected between 2018 and 2020, as mentioned in the manuscript.

References

  1. Line 340. Please review this reference. The title is repeated. The information of the publisher or the web page where it was published must be included.
  2. Line 345. Please, delete white space before de word Therapy
  3. Line 349. Please, replace CART with cART
  4. Line 352. Please, delete white space before de word Therapy
  5. Line 357. Please, replace CART with cART
  6. Please, review and rectified as correspond the whole reference.
  7. Line 377. Please, delete white space after de word Disease
  8. Line 383. Please, replace CART with cART
  9. Line 391. Please, replace CART with cART
  10. Line 400. Please, delete unnecessary white space before doi
  11. Line 442. Please, delete unnecessary white space before doi
  12. Line 458. Add space after de word HIV. Delete white space

Response: We appreciate the recommendations and all requested changes were done.

Supplementary Material

Tables, figures, or other supplementary material should appear in the order in which they are cited in the text of the manuscript.

Supplementary Table S1. Please, replace letter b with a

Response: We appreciate the recommendations and the indicated substitutions were done.

Reviewer 2 Report

In the present paper, Dos Santos et al, conduct a flow-cytometric analysis on HIV-infected immunological non responders (INR) and immunological responders (IR) on cART. Subjects were evalauted following 18 months of cART and the immunologic response was defined as a CD4+ T cell count > 500/ul.

The authors found:

- a higher proportion of INR in subjects with a low CD4 T-cell nadir

- higher levels of CD4+ T-cell activation in INR

- lower levels of RTE and naive CD4+ T-cells in INR

The study is well conducted and the  paper is well written. The results presented are in line with previous reports on the topic and as such do not add significantly to existing evidence.

I have the following comments to make:

- the overall number of subjects enrolled is small

- some subjects appear to be treated with older cART regimens; does this reflect local/national guidelines and/or drug availability? This should be commented in the text.

- aside from CD4+ nadir, authors should include how many subjects had AIDS defining events

- 18 months is a relatively short time-frame to assess immune recovery; have the authors attempted to analyze subjects at a later time-point? Further, were all subjects virally-suppressed at the time of analysis?

- can the authors comment their finding of comparable CD8+ T-cell activation between the two groups?

Author Response

Reviewer #2

General Comments

In the present paper, Dos Santos et al, conduct a flow-cytometric analysis on HIV-infected immunological non responders (INR) and immunological responders (IR) on cART. Subjects were evaluated following 18 months of cART and the immunologic response was defined as a CD4+ T cell count > 500/ul.

The authors found:

- a higher proportion of INR in subjects with a low CD4 T-cell nadir

- higher levels of CD4+ T-cell activation in INR

- lower levels of RTE and naive CD4+ T-cells in INR

The study is well conducted and the paper is well written. The results presented are in line with previous reports on the topic and as such do not add significantly to existing evidence.

I have the following comments to make:

- the overall number of subjects enrolled is small

Response: We understand your consideration, but other functional studies similar to this one have been also conducted  with small number of patients in the groups [1–3]. In flow cytometry analysis, it is common to use sample size around 40-60 subjects since this overall number provides sufficiently robust data to guarantee the study reliability. Moreover, the study population is well characterized within the inclusion criteria to maximally avoid any external bias, increasing the data reliability. Therefore, we believe that increasing the number of subjects would not significantly change the statistical results.

- some subjects appear to be treated with older cART regimens; does this reflect local/national guidelines and/or drug availability? This should be commented in the text.

Response: The ART regimens used by the patients in this study are based on World Health Organization guidelines (OMS) [4] and Brazilian recommendations [5]. Most of the patients (~80%) are treated by first-line regimens, which consist of tenofovir+lamivudine+dolutegravir or efavirenz (TDF+3TC+DTG or EFZ). However, if any of these antiretrovirals is not available, other drugs also recommended by WHO can be incorporated into the treatment, as alternative regimens. We appreciate your suggestion, and we have inserted this information in the manuscript for a better understanding of the study population.

- aside from CD4+ nadir, authors should include how many subjects had AIDS defining events

Response: To be recruited and included in the study for immunological recovery investigation, patients needed to meet some inclusion criteria, including prolonged virological suppression and absence of AIDS defining events.

- 18 months is a relatively short time-frame to assess immune recovery; have the authors attempted to analyze subjects at a later time-point? Further, were all subjects virally-suppressed at the time of analysis?

Response: We understand and appreciate your concern, but our immune classification was based on an adaptation of Cenderello et al. 2016, which proposes a classification after 18 to 36 months of treatment [6]. In addition, it has been demonstrated that the greatest gain in CD4+ T cells occurs during the first year of ART after virological suppression (within ~6 months of treatment)  [5,7]. Thus, the time used for patient classification (18 months: 6 months for virological suppression + 12 months for immunological recovery) is within an acceptable range. Furthermore, during study, all patients must have undetectable plasma viral load since it is the main inclusion criteria for immunological recovery investigation.

- can the authors comment their finding of comparable CD8+ T-cell activation between the two groups?

Response: Some studies have shown that CD8+ T cell activation (CD38+HLD-DR+) is significantly higher during pre-ART, and the virological suppression, resulted from treatment adherence, is responsible for decreasing this immune activation [8–10]. Although, residual viral replication naturally occurs in ART-treated HIV-positive patients, the subjects of this study showed undetectable plasma viral load, which decrease activation CD8+ T cells. However, it has been demonstrated that immune activation can persist by CD4+ T cell activation and turnover during ART even in virological suppression [11,12] corroborating with our results. Another study published by our group demonstrated the same pattern of CD8+ T cell activation between both IR and INR groups [13]

References

  1. Horta, A.; Nobrega, C.; Amorim-Machado, P.; Coutinho-Teixeira, V.; Barreira-Silva, P.; Boavida, S.; Costa, P.; Sarmento-Castro, R.; Castro, A.G.; Correia-Neves, M. Poor Immune Reconstitution in HIV-Infected Patients Associates with High Percentage of Regulatory CD4+ T Cells. PLoS One 2013, 8, 1–7, doi:10.1371/journal.pone.0057336.
  2. Gelpi, M.; Hartling, H.J.; Thorsteinsson, K.; Gerstoft, J.; Ullum, H.; Nielsen, S.D. Immune Recovery in Acute and Chronic HIV Infection and the Impact of Thymic Stromal Lymphopoietin. BMC Infect. Dis. 2016, 16, doi:10.1186/s12879-016-1930-3.
  3. Younes, S.A.; Talla, A.; Ribeiro, S.P.; Saidakova, E. V.; Korolevskaya, L.B.; Shmagel, K. V.; Shive, C.L.; Freeman, M.L.; Panigrahi, S.; Zweig, S.; et al. Cycling CD4+ T Cells in HIV-Infected Immune Nonresponders Have Mitochondrial Dysfunction. J. Clin. Invest. 2018, 128, 5083–5094, doi:10.1172/JCI120245.
  4. World Health Organization (WHO) Consolidated Guidelines on the Use of Antiretroviral Drugs for Treating and Preventing HIV Infection: Recommendations for a Public Health Approach. World Health Organization. WHO 2016, doi:10.1097/00022744-199706000-00003.
  5. Ministério da Saúde Protocolo Clínico e Diretrizes Terapêuticas Para Manejo Da Infecção Pelo HIV Em Adultos; 2018; Vol. 88; ISBN 9788533426405.
  6. Cenderello, G.; De Maria, A. Discordant Responses to CART in HIV-1 Patients in the Era of High Potency Antiretroviral Drugs: Clinical Evaluation, Classification, Management Prospects. Expert Rev. Anti. Infect. Ther. 2016, 14, 29–40, doi:10.1586/14787210.2016.1106937.
  7. Corbeau, P.; Reynes, J. Review Article Immune Reconstitution under Antiretroviral Therapy : The New Challenge in HIV-1 Infection. Therapy 2011, 117, 5582–5590, doi:10.1182/blood-2010-12-322453.
  8. Lok, J.J.; Hunt, P.W.; Collier, A.C.; Benson, C.A.; Witt, M.D.; Luque, A.E.; Deeks, S.G.; Bosch, R.J. The Impact of Age on the Prognostic Capacity of CD8+ T-Cell Activation during Suppressive Antiretroviral Therapy. AIDS 2013, 27, 2101–2110, doi:10.1097/QAD.0b013e32836191b1.
  9. D’Amico, R.; Yang, Y.; Mildvan, D.; Evans, S.R.; Schnizlein-Bick, C.T.; Hafner, R.; Webb, N.; Basar, M.; Zackin, R.; Jacobson, M.A. Lower CD4+ T Lymphocyte Nadirs May Indicate Limited Immune Reconstitution in HIV-1 Infected Individuals on Potent Antiretroviral Therapy: Analysis of Immunophenotypic Marker Results of AACTG 5067. J. Clin. Immunol. 2005, 25, 106–115, doi:10.1007/s10875-005-2816-0.
  10. Warren, J.A.; Clutton, G.; Goonetilleke, N. Harnessing CD8+ T Cells under HIV Antiretroviral Therapy. Front. Immunol. 2019, 10, 1–14, doi:10.3389/fimmu.2019.00291.
  11. Piconi, S.; Trabattoni, D.; Gori, A.; Parisotto, S.; Magni, C.; Meraviglia, P.; Bandera, A.; Capetti, A.; Rizzardini, G.; Clerici, M. Immune Activation, Apoptosis, and Treg Activity Are Associated with Persistently Reduced CD4+ T-Cell Counts during Antiretroviral Therapy. Aids 2010, 24, 1991–2000, doi:10.1097/QAD.0b013e32833c93ce.
  12. Yang, X.; Zhang, T.; Su, B.; Zhang, X.; Liu, Y.; Wu, H. Incomplete Immune Reconstitution in HIV / AIDS Patients on Antiretroviral Therapy : Challenges of Immunological Non-Responders. 2020, 1–16, doi:10.1002/JLB.4MR1019-189R.
  13. Carvalho-Silva, W.H.V.; Andrade-Santos, J.L.; Souto, F.O.; Coelho, A.V.C.; Crovella, S.; Guimarães, R.L. Immunological Recovery Failure in CART-Treated HIV-Positive Patients Is Associated with Reduced Thymic Output and RTE CD4+ T Cell Death by Pyroptosis. J. Leukoc. Biol. 2020, 107, 85–94, doi:10.1002/JLB.4A0919-235R.

Round 2

Reviewer 2 Report

Tha authors have addressed my concerns.